# EPOpt: Learning Robust Neural Network Policies Using Model Ensembles

**Aravind Rajeswaran**[1], **Sarvjeet Ghotra**[2], **Balaraman Ravindran**[3], **Sergey Levine**[4]
`aravraj@cs.washington.edu, sarvjeet.13it236@nitk.edu.in,`
`ravi@cse.iitm.ac.in, svlevine@eecs.berkeley.edu`
[1] University of Washington Seattle
[2] NITK Surathkal
[3] Indian Institute of Technology Madras
[4] University of California Berkeley

## Abstract

Sample complexity and safety are major challenges when learning policies with reinforcement learning for real-world tasks, especially when the policies are represented using rich function approximators like deep neural networks. Model-based methods where the real-world target domain is approximated using a simulated source domain provide an avenue to tackle the above challenges by augmenting real data with simulated data. However, discrepancies between the simulated source domain and the target domain pose a challenge for simulated training. We introduce the EPOpt algorithm, which uses an ensemble of simulated source domains and a form of adversarial training to learn policies that are robust and generalize to a broad range of possible target domains, including unmodeled effects. Further, the probability distribution over source domains in the ensemble can be adapted using data from target domain and approximate Bayesian methods, to progressively make it a better approximation. Thus, learning on a model ensemble, along with source domain adaptation, provides the benefit of both robustness and learning/adaptation.

## 1 Introduction

Reinforcement learning with powerful function approximators like deep neural networks (deep RL) has recently demonstrated remarkable success in a wide range of tasks like games (Mnih et al., 2015; Silver et al., 2016), simulated control problems (Lillicrap et al., 2015; Mordatch et al., 2015b), and graphics (Peng et al., 2016). However, high sample complexity is a major barrier for directly applying model-free deep RL methods for physical control tasks. Model-free algorithms like Q-learning, actor-critic, and policy gradients are known to suffer from long learning times (Kakade, 2003), which is compounded when used in conjunction with expressive function approximators like deep neural networks (DNNs). The challenge of gathering samples from the real world is further exacerbated by issues of safety for the agent and environment, since sampling with partially learned policies could be unstable (García & Fernández, 2015). Thus, model-free deep RL methods often require a prohibitively large numbers of potentially dangerous samples for physical control tasks.

Model-based methods, where the real-world target domain is approximated with a simulated source domain, provide an avenue to tackle the above challenges by learning policies using simulated data. The principal challenge with simulated training is the systematic discrepancy between source and target domains, and therefore, methods that compensate for systematic discrepancies (modeling errors) are needed to transfer results from simulations to real world using RL. We show that the impact of such discrepancies can be mitigated through two key ideas: (1) training on an ensemble of models in an adversarial fashion to learn policies that are robust to parametric model errors, as well as to unmodeled effects; and (2) adaptation of the source domain ensemble using data from the target domain to progressively make it a better approximation. This can be viewed either as an instance of model-based Bayesian RL (Ghavamzadeh et al., 2015); or as transfer learning from a collection of simulated source domains to a real-world target domain (Taylor & Stone, 2009). While a number of model-free RL algorithms have been proposed (see, e.g., Duan et al. (2016) for a survey), their high sample complexity demands use of a simulator, effectively making them model-based. We

show in our experiments that such methods learn policies which are highly optimized for the specific models used in the simulator, but are brittle under model mismatch. This is not surprising, since deep networks are remarkably proficient at exploiting any systematic regularities in a simulator. Addressing robustness of DNN-policies is particularly important to transfer their success from simulated tasks to physical systems.

In this paper, we propose the Ensemble Policy Optimization (EPOpt$-\epsilon$) algorithm for finding policies that are robust to model mismatch. In line with model-based Bayesian RL, we learn a policy for the target domain by alternating between two phases: (i) given a source (model) distribution (i.e. ensemble of models), find a robust policy that is competent for the whole distribution; (ii) gather data from the target domain using said robust policy, and adapt the source distribution. EPOpt uses an ensemble of models sampled from the source distribution, and a form of adversarial training to learn robust policies that generalize to a broad range of models. By robust, we mean insensitivity to parametric model errors and broadly competent performance for *direct-transfer* (also referred to as *jumpstart* like in Taylor & Stone (2009)). Direct-transfer performance refers to the average initial performance (return) in the target domain, without any direct training on the target domain. By adversarial training, we mean that model instances on which the policy performs poorly in the source distribution are sampled more often in order to encourage learning of policies that perform well for a wide range of model instances. This is in contrast to methods which learn highly optimized policies for specific model instances, but brittle under model perturbations. In our experiments, we did not observe significant loss in performance by requiring the policy to work on multiple models (for example, through adopting a more conservative strategy). Further, we show that policies learned using EPOpt are robust even to effects not modeled in the source domain. Such unmodeled effects are a major issue when transferring from simulation to the real world. For the model adaptation step (ii), we present a simple method using approximate Bayesian updates, which progressively makes the source distribution a better approximation of the target domain. We evaluate the proposed methods on the hopper (12 dimensional state space; 3 dimensional action space) and half-cheetah (18 dimensional state space; 6 dimensional action space) benchmarks in MuJoCo. Our experimental results suggest that adversarial training on model ensembles produces robust policies which generalize better than policies trained on a single, maximum-likelihood model (of source distribution) alone.

## 2 PROBLEM FORMULATION

We consider parametrized Markov Decision Processes (MDPs), which are tuples of the form: $\mathcal{M}(p) \equiv< \mathcal{S}, \mathcal{A}, \mathcal{T}_p, \mathcal{R}_p, \gamma, S_{0,p} >$ where $\mathcal{S}$, $\mathcal{A}$ are (continuous) states and actions respectively; $\mathcal{T}_p$ $\mathcal{R}_p$, and $S_{0,p}$ are the state transition, reward function, and initial state distribution respectively, all parametrized by $p$; and $\gamma$ is the discount factor. Thus, we consider a set of MDPs with the same state and action spaces. Each MDP in this set could potentially have different transition functions, rewards, and initial state distributions. We use transition functions of the form $S_{t+1} \equiv \mathcal{T}_p(s_t, a_t)$ where $\mathcal{T}_p$ is a random process and $S_{t+1}$ is a random variable.

We distinguish between source and target MDPs using $\mathcal{M}$ and $\mathcal{W}$ respectively. We also refer to $\mathcal{M}$ and $\mathcal{W}$ as source and target domains respectively, as is common in the transfer learning set-up. Our objective is to learn the optimal policy for $\mathcal{W}$; and to do so, we have access to $\mathcal{M}(p)$. We assume that we have a distribution ($\mathcal{D}$) over the source domains (MDPs) generated by a distribution over the parameters $P \equiv \mathcal{P}(p)$ that capture our subjective belief about the parameters of $\mathcal{W}$. Let $\mathcal{P}$ be parametrized by $\psi$ (e.g. mean, standard deviation). For example, $\mathcal{M}$ could be a hopping task with reward proportional to hopping velocity and falling down corresponds to a terminal state. For this task, $p$ could correspond to parameters like torso mass, ground friction, and damping in joints, all of which affect the dynamics. Ideally, we would like the target domain to be in the model class, i.e. $\{\exists p \mid \mathcal{M}(p) = \mathcal{W}\}$. However, in practice, there are likely to be unmodeled effects, and we analyze this setting in our experiments. We wish to learn a policy $\pi_\theta^*(s)$ that performs well for all $\mathcal{M} \sim \mathcal{D}$. Note that this robust policy does not have an explicit dependence on $p$, and we require it to perform well without knowledge of $p$.

## 3 LEARNING PROTOCOL AND EPOPT ALGORITHM

We follow the round-based learning protocol of Bayesian model-based RL. We use the term *rounds* when interacting with the target domain, and *episode* when performing rollouts with the simulator. In each round, we interact with the target domain after computing the robust policy on the current (i.e.

posterior) simulated source distribution. Following this, we update the source distribution using data from the target domain collected by executing the robust policy. Thus, in round $i$, we update two sets of parameters: $\theta_i$, the parameters of the robust policy (neural network); and $\psi_i$, the parameters of the source distribution. The two key steps in this procedure are finding a robust policy given a source distribution; and updating the source distribution using data from the target domain. In this section, we present our approach for both of these steps.

## 3.1 ROBUST POLICY SEARCH

We introduce the EPOpt algorithm for finding a robust policy using the source distribution. EPOpt is a policy gradient based meta-algorithm which uses batch policy optimization methods as a subroutine. Batch policy optimization algorithms (Williams, 1992; Kakade, 2001; Schulman et al., 2015) collect a batch of trajectories by rolling out the current policy, and use the trajectories to make a policy update. The basic structure of EPOpt is to sample a collection of models from the source distribution, sample trajectories from each of these models, and make a gradient update based on a subset of sampled trajectories. We first define evaluation metrics for the parametrized policy, $\pi_\theta$:

$$\eta_{\mathcal{M}}(\theta, p) = \mathbb{E}_{\tilde{\tau}} \left[ \sum_{t=0}^{T-1} \gamma^t r_t(s_t, a_t) \,\middle|\, p \right],  \tag{1}$$

$$\eta_{\mathcal{D}}(\theta) = \mathbb{E}_{p \sim \mathcal{P}} \left[ \eta_{\mathcal{M}}(\theta, p) \right] = \mathbb{E}_{p \sim \mathcal{P}} \left[ \mathbb{E}_{\hat{\tau}} \left[ \sum_{t=0}^{T-1} \gamma^t r_t(s_t, a_t) \,\middle|\, p \right] \right] = \mathbb{E}_{\tau} \left[ \sum_{t=0}^{T-1} \gamma^t r_t(s_t, a_t) \right].$$

In (1), $\eta_{\mathcal{M}}(\theta, p)$ is the evaluation of $\pi_\theta$ on the model $\mathcal{M}(p)$, with $\tilde{\tau}$ being trajectories generated by $\mathcal{M}(p)$ and $\pi_\theta$: $\tilde{\tau} = \{s_t, a_t, r_t\}_{t=0}^{T}$ where $s_{t+1} \sim \mathcal{T}_p(s_t, a_t)$, $s_0 \sim S_{0,p}$, $r_t \sim \mathcal{R}_p(s_t, a_t)$, and $a_t \sim \pi_\theta(s_t)$. Similarly, $\eta_{\mathcal{D}}(\theta)$ is the evaluation of $\pi_\theta$ over the source domain distribution. The corresponding expectation is over trajectories $\tau$ generated by $\mathcal{D}$ and $\pi_\theta$: $\tau = \{s_t, a_t, r_t\}_{t=0}^{T}$, where $s_{t+1} \sim \mathcal{T}_{p_t}(s_t, a_t)$, $p_{t+1} = p_t$, $s_0 \sim S_{0,p_0}$, $r_t \sim \mathcal{R}_{p_t}(s_t, a_t)$, $a_t \sim \pi_\theta(s_t)$, and $p_0 \sim \mathcal{P}$. With this modified notation of trajectories, batch policy optimization can be invoked for policy search.

Optimizing $\eta_{\mathcal{D}}$ allows us to learn a policy that performs best in expectation over models in the source domain distribution. However, this does not necessarily lead to a robust policy, since there could be high variability in performance for different models in the distribution. To explicitly seek a robust policy, we use a softer version of max-min objective suggested in robust control, and optimize for the conditional value at risk (CVaR) (Tamar et al., 2015):

$$\max_{\theta, y} \int_{\mathcal{F}(\theta)} \eta_{\mathcal{M}}(\theta, p) \mathcal{P}(p) dp \qquad s.t. \quad \mathbb{P}\left( \eta_{\mathcal{M}}(\theta, P) \leq y \right) = \epsilon,  \tag{2}$$

where $\mathcal{F}(\theta) = \{p \mid \eta_{\mathcal{M}}(\theta, p) \leq y\}$ is the set of parameters corresponding to models that produce the worst $\epsilon$ percentile of returns, and provides the limit for the integral; $\eta_{\mathcal{M}}(\theta, P)$ is the random variable of returns, which is induced by the distribution over model parameters; and $\epsilon$ is a hyperparameter which governs the level of relaxation from max-min objective. The interpretation is that (2) maximizes the expected return for the worst $\epsilon$-percentile of MDPs in the source domain distribution. We adapt the previous policy gradient formulation to approximately optimize the objective in (2). The resulting algorithm, which we call EPOpt-$\epsilon$, generalizes learning a policy using an ensemble of source MDPs which are sampled from a source domain distribution.

In Algorithm 1, $R(\tau_k) \equiv \sum_{t=0}^{T-1} \gamma^t r_{t,k}$ denotes the discounted return obtained in trajectory sample $\tau_k$. In line 7, we compute the $\epsilon-$percentile value of returns from the $N$ trajectories. In line 8, we find the subset of sampled trajectories which have returns lower than $Q_\epsilon$. Line 9 calls one step of an underlying batch policy optimization subroutine on the subset of trajectories from line 8. For the CVaR objective, it is important to use a good baseline for the value function. Tamar et al. (2015) show that without a baseline, the resulting policy gradient is biased and not consistent. We use a linear function as the baseline with a time varying feature vector to approximate the value function, similar to Duan et al. (2016). The parameters of the baseline are estimated using only the subset of trajectories with return less than $Q_\epsilon$. We found that this approach led to empirically good results.

For small values of $\epsilon$, we observed that using the sub-sampling step from the beginning led to unstable learning. Policy gradient methods adjust parameters of policy to increase probability of trajectories

---

**Algorithm 1:** EPOpt–$\epsilon$ for Robust Policy Search

---

1  **Input:** $\psi$, $\theta_0$, $niter$, $N$, $\epsilon$
2  **for** *iteration* $i = 0, 1, 2, \ldots niter$ **do**
3 **for** $k = 1, 2, \ldots N$ **do**
4 sample model parameters $p_k \sim \mathcal{P}_\psi$
5 sample a trajectory $\tau_k = \{s_t, a_t, r_t, s_{t+1}\}_{t=0}^{T-1}$ from $\mathcal{M}(p_k)$ using policy $\pi(\theta_i)$
6 **end**
7 compute $Q_\epsilon = \epsilon$ percentile of $\{R(\tau_k)\}_{k=1}^N$
8 select sub-set $\mathbb{T} = \{\tau_k : R(\tau_k) \leq Q_\epsilon\}$
9 Update policy: $\theta_{i+1} = \text{BatchPolOpt}(\theta_i, \mathbb{T})$
10 **end**

---

with high returns and reduce probability of poor trajectories. EPOpt$-\epsilon$ due to the sub-sampling step emphasizes penalizing poor trajectories more. This might constrain the initial exploration needed to find good trajectories. Thus, we initially use a setting of $\epsilon = 1$ for few iterations before setting epsilon to the desired value. This corresponds to exploring initially to find promising trajectories and rapidly reducing probability of trajectories that do not generalize.

### 3.2 Adapting the source domain distribution

In line with model-based Bayesian RL, we can adapt the ensemble distribution after observing trajectory data from the target domain. The Bayesian update can be written as:

$$\mathbb{P}(P|\tau_k) = \frac{1}{Z} \times \mathbb{P}(\tau_k|P) \times \mathbb{P}(P) \quad = \frac{1}{Z} \times \prod_{t=0}^{T-1} \mathbb{P}(S_{t+1} = s_{t+1}^{(k)}|s_t^{(k)}, a_t^{(k)}, p) \times \mathbb{P}(P = p), \quad (3)$$

where $\frac{1}{Z}$ is the partition function (normalization) required to make the probabilities sum to 1, $S_{t+1}$ is the random variable representing the next state, and $\left(s_t^{(k)}, a_t^{(k)}, s_{t+1}^{(k)}\right)_{t=0}^{T}$ are data observed along trajectory $\tau_k$. We try to explain the target trajectory using the stochasticity in the state-transition function, which also models sensor errors. This provides the following expression for the likelihood:

$$\mathbb{P}(S_{t+1}|s_t, a_t, p) \equiv \mathcal{T}_p(s_t, a_t). \quad (4)$$

We follow a sampling based approach to calculate the posterior, by sampling a set of model parameters: $p_i = [p_1, p_2, \ldots, p_M]$ from a sampling distribution, $\mathbb{P}_S(p_i)$. Consequently, using Bayes rule and importance sampling, we have:

$$\mathbb{P}(p_i|\tau_k) \propto \mathcal{L}(\tau_k|p_i) \times \frac{\mathbb{P}_P(p_i)}{\mathbb{P}_S(p_i)}, \quad (5)$$

where $\mathbb{P}_P(p_i)$ is the probability of drawing $p_i$ from the prior distribution; and $\mathcal{L}(\tau_k|p_i)$ is the likelihood of generating the observed trajectory with model parameters $p_i$. The weighted samples from the posterior can be used to estimate a parametric model, as we do in this paper. Alternatively, one could approximate the continuous probability distribution using discrete weighted samples like in case of particle filters. In cases where the prior has very low probability density in certain parts of the parameter space, it might be advantageous to choose a sampling distribution different from the prior. The likelihood can be factored using the Markov property as: $\mathcal{L}(\tau_k|p_i) = \prod_t \mathbb{P}(S_{t+1} = s_{t+1}^{(k)}|s_t^{(k)}, a_t^{(k)}, p_i)$. This simple model adaptation rule allows us to illustrate the utility of EPOpt for robust policy search, as well as its integration with model adaptation to learn policies in cases where the target model could be very different from the initially assumed distribution.

## 4 Experiments

We evaluated the proposed EPOpt-$\epsilon$ algorithm on the 2D hopper (Erez et al., 2011) and half-cheetah (Wawrzynski, 2009) benchmarks using the MuJoCo physics simulator (Todorov et al., 2012).[1] Both tasks involve complex second order dynamics and direct torque control. Underactuation,

---

[1]Supplementary video: https://youtu.be/w1YJ9vwaoto

high dimensionality, and contact discontinuities make these tasks challenging reinforcement learning benchmarks. These challenges when coupled with systematic parameter discrepancies can quickly degrade the performance of policies and make them unstable, as we show in the experiments. The batch policy optimization sub-routine is implemented using TRPO. We parametrize the stochastic policy using the scheme presented in Schulman et al. (2015). The policy is represented with a Gaussian distribution, the mean of which is parametrized using a neural network with two hidden layers. Each hidden layer has 64 units, with a *tanh* non-linearity, and the final output layer is made of linear units. Normally distributed independent random variables are added to the output of this neural network, and we also learn the standard deviation of their distributions. Our experiments are aimed at answering the following questions:

1. How does the performance of standard policy search methods (like TRPO) degrade in the presence of systematic physical differences between the training and test domains, as might be the case when training in simulation and testing in the real world?

2. Does training on a distribution of models with EPOpt improve the performance of the policy when tested under various model discrepancies, and how much does ensemble training degrade overall performance (e.g. due to acquiring a more conservative strategy)?

3. How does the robustness of the policy to physical parameter discrepancies change when using the robust EPOpt-$\epsilon$ variant of our method?

4. Can EPOpt learn policies that are robust to unmodeled effects – that is, discrepancies in physical parameters between source and target domains that *do not* vary in the source domain ensemble?

5. When the initial model ensemble differs substantially from the target domain, can the ensemble be adapted efficiently, and how much data from the target domain is required for this?

In all the comparisons, *performance* refers to the average undiscounted return per trajectory or episode (we consider finite horizon episodic problems). In addition to the previously defined performance, we also use the 10$^{th}$ percentile of the return distribution as a proxy for the worst-case return.

## 4.1 COMPARISON TO STANDARD POLICY SEARCH

In Figure 1, we evaluate the performance of standard TRPO and EPOpt($\epsilon = 0.1$) on the hopper task, in the presence of a simple parametric discrepancy in the physics of the system between the training (source) and test (target) domains. The plots show the performance of various policies on test domains with different torso mass. The first three plots show policies that are each trained on a single torso mass in the source domain, while the last plot illustrates the performance of EPOpt,

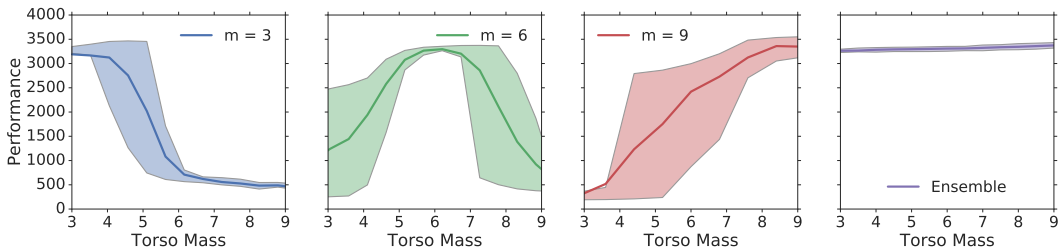

Figure 1: Performance of hopper policies when testing on target domains with different torso masses. The first three plots (blue, green, and red) show the performance of policies trained with TRPO on source domains with torso mass 3, 6, and 9, respectively (denoted by $m =$ in the legend). The rightmost plot shows the performance of EPOpt($\epsilon = 0.1$) trained on a Gaussian source distribution with mean mass $\mu = 6$ and standard deviation $\sigma = 1.5$. The shaded regions show the 10$^{th}$ and 90$^{th}$ percentile of the return distribution. Policies trained using traditional approaches on a single mass value are unstable for even slightly different masses, making the hopper fall over when trying to move forward. In contrast, the EPOpt policy is stable and achieves a high level of performance on the entire range of masses considered. Further, the EPOpt policy does not suffer from degradation in performance as a consequence of adopting a more robust policy.

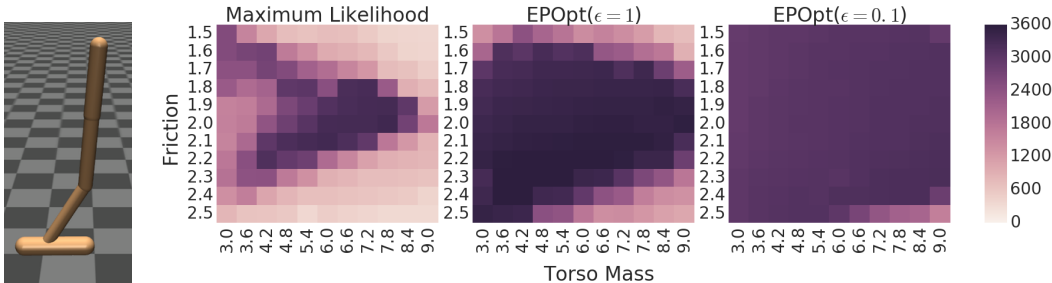

Figure 2: On the left, is an illustration of the simulated 2D hopper task studied in this paper. On right, we depict the performance of policies for various model instances of the hopper task. The performance is depicted as a heat map for various model configurations, parameters of which are given in the x and y axis. The adversarially trained policy, EPOpt($\epsilon = 0.1$), is observed to generalize to a wider range of models and is more robust.

which is trained on a Gaussian mass distribution. The results show that no single torso mass value produces a policy that is successful in all target domains. However, the EPOpt policy succeeds almost uniformly for all tested mass values. Furthermore, the results show that there is almost no degradation in the performance of EPOpt for any mass setting, suggesting that the EPOpt policy does not suffer substantially from adopting a more robust strategy.

## 4.2 ANALYSIS OF ROBUSTNESS

Next, we analyze the robustness of policies trained using EPOpt on the hopper domain. For this analysis, we construct a source distribution which varies four different physical parameters: torso mass, ground friction, foot joint damping, and joint inertia (armature). This distribution is presented in Table 1. Using this source distribution, we compare between three different methods: (1) standard policy search (TRPO) trained on a single model corresponding to the mean parameters in Table 1; (2) EPOpt($\epsilon = 1$) trained on the source distribution; (3) EPOpt($\epsilon = 0.1$) – i.e. the adversarially trained policy, again trained on the previously described source distribution. The aim of the comparison is to study direct-transfer performance, similar to the robustness evaluations common in robust controller design (Zhou et al., 1996). Hence, we learn a policy using each of the methods, and then test policies on different model instances (i.e. different combinations of physical parameters) without any adaptation. The results of this comparison are summarized in Figure 2, where we present the performance of the policy for testing conditions corresponding to different torso mass and friction values, which we found to have the most pronounced impact on performance. The results indicate that EPOpt($\epsilon = 0.1$) produces highly robust policies. A similar analysis for the 10th percentile of the return distribution (softer version of worst-case performance), the half-cheetah task, and different $\epsilon$ settings are presented in the appendix.

Table 1: Initial source domain distribution

| **Hopper** | $\mu$ | $\sigma$ | low | high |
|---|---|---|---|---|
| mass | 6.0 | 1.5 | 3.0 | 9.0 |
| ground friction | 2.0 | 0.25 | 1.5 | 2.5 |
| joint damping | 2.5 | 1.0 | 1.0 | 4.0 |
| armature | 1.0 | 0.25 | 0.5 | 1.5 |
| **Half-Cheetah** | $\mu$ | $\sigma$ | low | high |
| mass | 6.0 | 1.5 | 3.0 | 9.0 |
| ground friction | 0.5 | 0.1 | 0.3 | 0.7 |
| joint damping | 1.5 | 0.5 | 0.5 | 2.5 |
| armature | 0.125 | 0.04 | 0.05 | 0.2 |

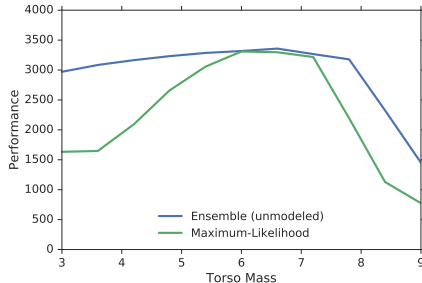

Figure 3: Comparison between policies trained on a fixed *maximum-likelihood* model with mass (6), and an ensemble where all models have the same mass (6) and other parameters varying as described in Table 1.

### 4.3    ROBUSTNESS TO UNMODELED EFFECTS

To analyze the robustness to unmodeled effects, our next experiment considers the setting where the source domain distribution is obtained by varying friction, damping, and armature as in Table 1, but does not consider a distribution over torso mass. Specifically, all models in the source domain distribution have the same torso mass (value of 6), but we will evaluate the policy trained on this distribution on target domains where the torso mass is different. Figure 3 indicates that the EPOpt($\epsilon = 0.1$) policy is robust to a broad range of torso masses even when its variation is not considered. However, as expected, this policy is not as robust as the case when mass is also modeled as part of the source domain distribution.

### 4.4    MODEL ADAPTATION

The preceding experiments show that EPOpt can find robust policies, but the source distribution in these experiments was chosen to be broad enough such that the target domain is not too far from high-density regions of the distribution. However, for real-world problems, we might not have the domain knowledge to identify a good source distribution in advance. In such settings, model (source) adaptation allows us to change the parameters of the source distribution using data gathered from the target domain. Additionally, model adaptation is helpful when the parameters of the target domain could change over time, for example due to wear and tear in a physical system. To illustrate model adaptation, we performed an experiment where the target domain was very far from the high density regions of the initial source distribution, as depicted in Figure 4(a). In this experiment, the source distribution varies the torso mass and ground friction. We observe that progressively, the source distribution becomes a better approximation of the target domain and consequently the performance improves. In this case, since we followed a sampling based approach, we used a uniform sampling distribution, and weighted each sample with the importance weight as described in Section 3.2. Eventually, after 10 iterations, the source domain distribution is able to accurately match the target domain. Figure 4(b) depicts the learning curve, and we see that a robust policy with return more than 2500, which roughly corresponds to a situation where the hopper is able to move forward without falling down for the duration of the episode, can be discovered with just 5 trajectories from the target domain. Subsequently, the policy improves near monotonically, and EPOpt finds a good policy with just 11 episodes worth of data from the target domain. In contrast, to achieve the same level of performance on the target domain, completely model-free methods like TRPO would require more than $2 \times 10^4$ trajectories when the neural network parameters are initialized randomly.

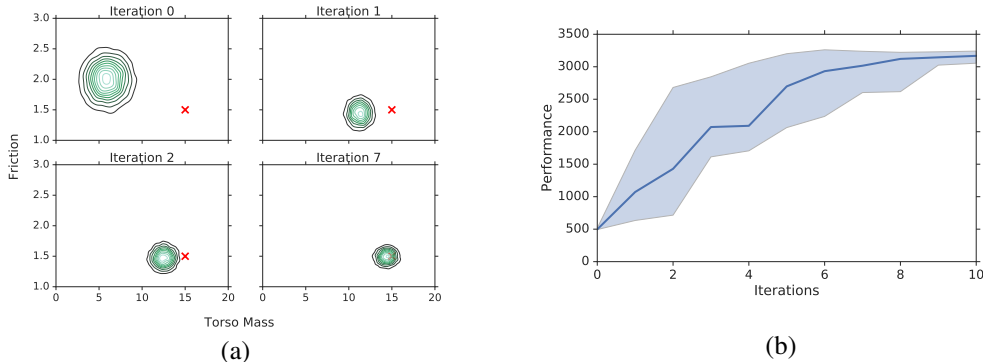

(a)                                                                              (b)

Figure 4: (a) Visualizes the source distribution during model adaptation on the hopper task, where mass and friction coefficient are varied in the source domain. The red cross indicates the unknown parameters of the target domain. The contours in the plot indicate the distribution over models (we assume a Gaussian distribution). Lighter colors and more concentrated contour lines indicate regions of higher density. Each iteration corresponds to one round (episode) of interaction with the target domain. The high-density regions gradually move toward the true model, while maintaining probability mass over a range of parameters which can explain the behavior of target domain. Figure 4(b) presents the corresponding learning curve, where the shaded region describes the 10th and 90th percentiles of the performance distribution, and the solid line is the average performance.

## 5  RELATED WORK

Robust control is a branch of control theory which formally studies development of robust policies (Zhou et al., 1996; Nilim & Ghaoui, 2005; Lim et al., 2013). However, typically no distribution over source or target tasks is assumed, and a worst case analysis is performed. Most results from this field have been concentrated around linear systems or finite MDPs, which often cannot adequately model complexities of real-world tasks. The set-up of model-based Bayesian RL maintains a belief over models for decision making under uncertainty (Vlassis et al., 2012; Ghavamzadeh et al., 2015). In Bayesian RL, through interaction with the target domain, the uncertainty is reduced to find the correct or closest model. Application of this idea in its full general form is difficult, and requires either restrictive assumptions like finite MDPs (Poupart et al., 2006), gaussian dynamics (Ross et al., 2008), or task specific innovations. Previous methods have also suggested treating uncertain model parameters as unobserved state variables in a continuous POMDP framework, and solving the POMDP to get optimal exploration-exploitation trade-off (Duff, 2003; Porta et al., 2006). While this approach is general, and allows automatic learning of epistemic actions, extending such methods to large continuous control tasks like those considered in this paper is difficult.

Risk sensitive RL methods (Delage & Mannor, 2010; Tamar et al., 2015) have been proposed to act as a bridge between robust control and Bayesian RL. These approaches allow for using subjective model belief priors, prevent overly conservative policies, and enjoy some strong guarantees typically associated with robust control. However, their application in high dimensional continuous control tasks have not been sufficiently explored. We refer readers to García & Fernández (2015) for a survey of related risk sensitive RL methods in the context of robustness and safety.

Standard model-based control methods typically operate by finding a maximum-likelihood estimate of the target model (Ljung, 1998; Ross & Bagnell, 2012; Deisenroth et al., 2013), followed by policy optimization. Use of model ensembles to produce robust controllers was explored recently in robotics. Mordatch et al. (2015a) use a trajectory optimization approach and an ensemble with small finite set of models; whereas we follow a sampling based direct policy search approach over a continuous distribution of uncertain parameters, and also show domain adaptation. Sampling based approaches can be applied to complex models and discrete MDPs which cannot be planned through easily. Similarly, Wang et al. (2010) use an ensemble of models, but their goal is to optimize for average case performance as opposed to transferring to a target MDP. Wang et al. (2010) use a hand engineered policy class whose parameters are optimized with CMA-ES. EPOpt on the other hand can optimize expressive neural network policies directly. In addition, we show model adaptation, effectiveness of the sub-sampling step ($\epsilon < 1$ case), and robustness to unmodeled effects, all of which are important for transfering to a target MDP.

Learning of parametrized skills (da Silva et al., 2012) is also concerned with finding policies for a distribution of parametrized tasks. However, this is primarily geared towards situations where task parameters are revealed during test time. Our work is motivated by situations where target task parameters (e.g. friction) are unknown. A number of methods have also been suggested to reduce sample complexity when provided with either a baseline policy (Thomas et al., 2015; Kakade & Langford, 2002), expert demonstration (Levine & Koltun, 2013; Argall et al., 2009), or approximate simulator (Tamar et al., 2012; Abbeel et al., 2006). These are complimentary to our work, in the sense that our policy, which has good direct-transfer performance, can be used to sample from the target domain and other off-policy methods could be explored for policy improvement.

## 6  CONCLUSIONS AND FUTURE WORK

In this paper, we presented the EPOpt-$\epsilon$ algorithm for training robust policies on ensembles of source domains. Our method provides for training of robust policies, and supports an adversarial training regime designed to provide good direct-transfer performance. We also describe how our approach can be combined with Bayesian model adaptation to adapt the source domain ensemble to a target domain using a small amount of target domain experience. Our experimental results demonstrate that the ensemble approach provides for highly robust and generalizable policies in fairly complex simulated robotic tasks. Our experiments also demonstrate that Bayesian model adaptation can produce distributions over models that lead to better policies on the target domain than more standard maximum likelihood estimation, particularly in presence of unmodeled effects.

Although our method exhibits good generalization performance, the adaptation algorithm we use currently relies on sampling the parameter space, which is computationally intensive as the number of variable physical parameters increase. We observed that (adaptive) sampling from the prior leads to fast and reliable adaptation if the true model does not have very low probability in the prior. However, when this assumption breaks, we require a different sampling distribution which could produce samples from all regions of the parameter space. This is a general drawback of Bayesian adaptation methods. In future work, we plan to explore alternative sampling and parameterization schemes, including non-parametric distributions. An eventual end-goal would be to replace the physics simulator entirely with learned Bayesian neural network models, which could be adapted with limited data from the physical system. These models could be pre-trained using physics based simulators like MuJoCo to get a practical initialization of neural network parameters. Such representations are likely useful when dealing with high dimensional inputs like simulated vision from rendered images or tasks with complex dynamics like deformable bodies, which are needed to train highly generalizable policies that can successfully transfer to physical robots acting in the real world.

### ACKNOWLEDGMENTS

The authors would like to thank Emo Todorov, Sham Kakade, and students of Emo Todorov's research group for insightful comments about the work. The authors would also like to thank Emo Todorov for the MuJoCo simulator. Aravind Rajeswaran and Balaraman Ravindran acknowledge financial support from ILDS, IIT Madras.

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

# A APPENDIX

## A.1 DESCRIPTION OF SIMULATED ROBOTIC TASKS CONSIDERED IN THIS WORK

**Hopper:** The hopper task is to make a 2D planar hopper with three joints and 4 body parts hop forward as fast as possible (Erez et al., 2011). This problem has a 12 dimensional state space and a 3 dimensional action space that corresponds to torques at the joints. We construct the source domain by considering a distribution over 4 parameters: torso mass, ground friction, armature (inertia), and damping of foot.

**Half Cheetah:** The half-cheetah task (Wawrzynski, 2009) requires us to make a 2D cheetah with two legs run forward as fast as possible. The simulated robot has 8 body links with an 18 dimensional state space and a 6 dimensional action space that corresponds to joint torques. Again, we construct the source domain using a distribution over the following parameters: torso and head mass, ground friction, damping, and armature (inertia) of foot joints.

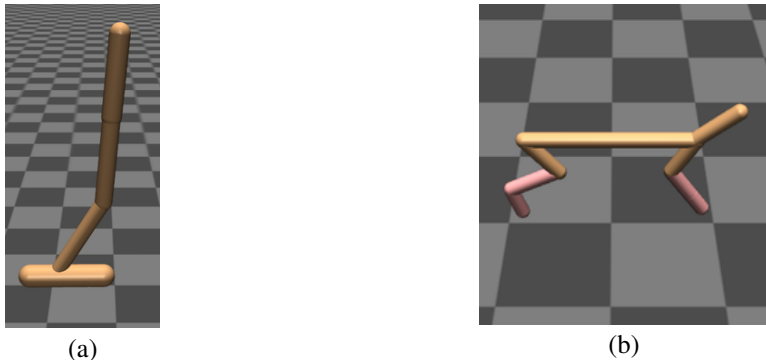

(a)                                    (b)

Figure 5: Illustrations of the 2D simulated robot models used in the experiments. The hopper (a) and half-cheetah (b) tasks present the challenges of under-actuation and contact discontinuities. These challenges when coupled with parameter uncertainties lead to dramatic degradation in the quality of policies when robustness is not explicitly considered.

A video demonstration of the trained policies on these tasks can be viewed here: Supplimenrary video ( `https://youtu.be/w1YJ9vwaoto` )

**Reward functions:** For both tasks, we used the standard reward functions implemented with OpenAI gym (Brockman et al., 2016), with minor modifications. The reward structure for hopper task is:

$$r(s, a) = v_x - 0.001||a||^2 + b,$$

where $s$ are the states comprising of joint positions and velocities; $a$ are the actions (controls); and $v_x$ is the forward velocity. $b$ is a bonus for being alive ($b = 1$). The episode terminates when $z_{\text{torso}} < 0.7$ or when $|\theta_y| < 0.2$ where $\theta_y$ is the forward pitch of the body.

For the cheetah task, we use the reward function:

$$r(s, a) = v_x - 0.1||a||^2 + b,$$

the alive bonus is 1 if head of cheetah is above $-0.25$ (relative to torso) and similarly episode terminates if the alive condition is violated.

Our implementation of the algorithms and environments are public in this repository to facilitate reproduction of results: `https://github.com/aravindr93/robustRL`

## A.2 HYPERPARAMETERS

1. Neural network architecture: We used a neural network with two hidden layers, each with 64 units and *tanh* non-linearity. The policy updates are implemented using TRPO.

2. Trust region size in TRPO: The maximum KL divergence between sucessive policy updates are constrained to be 0.01

3. Number and length of trajectory rollouts: In each iteration, we sample $N = 240$ models from the ensemble, one rollout is performed on each such model. This was implemented in parallel on multiple (6) CPUs. Each trajectory is of length $1000$ – same as the standard implimentations of these tasks in gym and rllab.

The results in Fig 1 and Fig 2 were generated after 150 and 200 iterations of TRPO respectively, with each iteration consisting of 240 trajectories as specified in (3) above.

## A.3   WORST-CASE ANALYSIS FOR HOPPER TASK

Figure 2 illustrates the performance of the three considered policies: viz. TRPO on mean parameters, EPOpt($\epsilon = 1$), and EPOpt($\epsilon = 0.1$). We similarly analyze the $10^{\text{th}}$ percentile of the return distribution as a proxy for worst-case analysis, which is important for a robust control policy (here, distribution of returns for a given model instance is due to variations in initial conditions). The corresponding results are presented below:

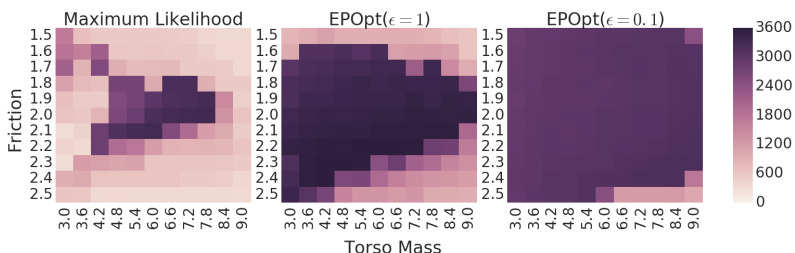

Figure 6: $10^{\text{th}}$ percentile of return distribution for the hopper task. EPOpt($\epsilon = 0.1$) clearly outperforms the other approaches. The $10^{\text{th}}$ of return distribution for EPOpt($\epsilon = 0.1$) also nearly overlaps with the expected return, indicating that the policies trained using EPOpt($\epsilon = 0.1$) are highly robust and reliable.

## A.4   ROBUSTNESS ANALYSIS FOR HALF-CHEETAH TASK

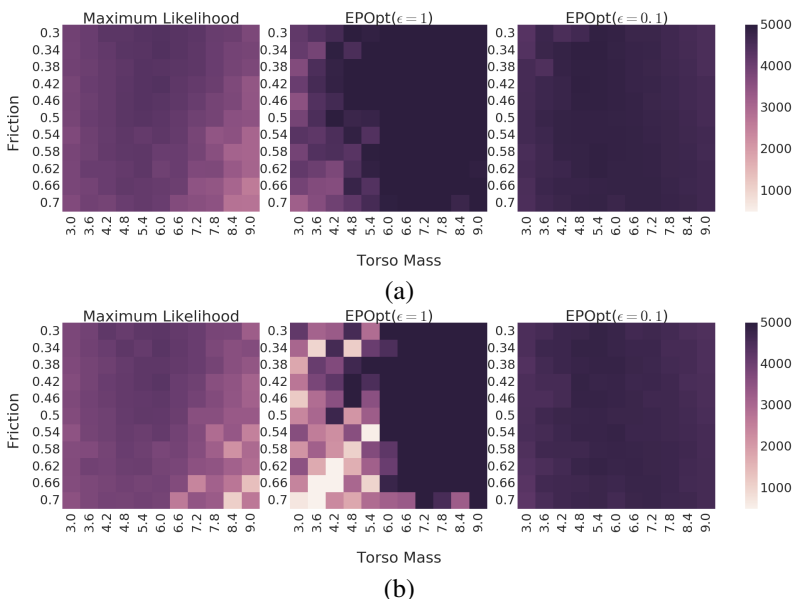

Figure 7: Performance of policies for various model instances for the half-cheetah domain, similar to Figure 2. Again, it is observed that the adversarial trained policy is robust and generalizes well to all models in the source distribution.

## A.5 DIFFERENT SETTINGS FOR $\epsilon$

Here, we analyze how different settings for $\epsilon$ influences the robustness of learned policies. The policies in this section have been trained for 200 iterations with 240 trajectory samples per iteration. Similar to the description in Section 3.1, the first 100 iterations use $\epsilon = 1$, and the final 100 iterations use the desired $\epsilon$. The source distribution is described in Table 1. We test the performance on a grid over the model parameters. Our results, summarized in Table 2, indicate that decreasing $\epsilon$ decreases the variance in performance, along with a small decrease in average performance, and hence enhances robustness.

Table 2: Performance statistics for different $\epsilon$ settings for the hopper task

| | | | **Performance (Return)** | | | | | |
|---|---|---|---|---|---|---|---|---|
| $\epsilon$ | **mean** | **std** | **Percentiles** | | | | | |
| | | | 5 | 10 | 25 | 50 | 75 | 90 |
| 0.05 | 2889 | 502 | 1662 | 2633 | 2841 | 2939 | 2966 | 3083 |
| 0.1 | 3063 | 579 | 1618 | 2848 | 3223 | 3286 | 3336 | 3396 |
| 0.2 | 3097 | 665 | 1527 | 1833 | 3259 | 3362 | 3423 | 3483 |
| 0.3 | 3121 | 706 | 1461 | 1635 | 3251 | 3395 | 3477 | 3513 |
| 0.4 | 3126 | 869 | 1013 | 1241 | 3114 | 3412 | 3504 | 3546 |
| 0.5 | 3122 | 1009 | 984 | 1196 | 1969 | 3430 | 3481 | 3567 |
| 0.75 | 3133 | 952 | 1005 | 1516 | 2187 | 3363 | 3486 | 3548 |
| 1.0 | 3224 | 1060 | 1198 | 1354 | 1928 | 3461 | 3557 | 3604 |
| Max-Lik | 1710 | 1140 | 352 | 414 | 646 | 1323 | 3088 | 3272 |

## A.6 IMPORTANCE OF BASELINE FOR BATCHPOLOPT

As described in Section 3.1, it is important to use a good baseline estimate for the value function for the batch policy optimization step. When optimizing for the expected return, we can interpret the baseline as a variance reduction technique. Intuitively, policy gradient methods adjust parameters of the policy to improve probability of trajectories in proportion to their performance. By using a baseline for the value function, we make updates that increase probability of trajectories that perform better than average and vice versa. In practice, this variance reduction is essential for getting policy gradients to work. For the CVaR case, Tamar et al. (2015) showed that without using a baseline, the policy gradient is biased. To study importance of the baseline, we first consider the case where we do not employ the adversarial sub-sampling step, and fix $\epsilon = 1$. We use a linear baseline with a time-varying feature vector as described in Section 3.1. Figure 8(a) depicts the learning curve for the source distribution in Table 1. The results indicate that use of a baseline is important to make policy gradients work well in practice.

Next, we turn to the case of $\epsilon < 1$. As mentioned in section 3.1, setting a low $\epsilon$ from the start leads to unstable learning. The adversarial nature encourages penalizing poor trajectories more, which constrains the initial exploration needed to find promising trajectories. Thus we will "pre-train" by using $\epsilon = 1$ for some iterations, before switching to the desired $\epsilon$ setting. From Figure 8(a), it is clear that pre-training without a baseline is unlikely to help, since the performance is poor. Thus, we use the following setup for comparison: for 100 iterations, EPOpt($\epsilon = 1$) is used with the baseline. Subsequently, we switch to EPOpt($\epsilon = 0.1$) and run for another 100 iterations, totaling 200 iterations. The results of this experiment are depicted in Figure 8(b). This result indicates that use of a baseline is crucial for the CVaR case, without which the performance degrades very quickly. We repeated the experiment with 100 iterations of pre-training with $\epsilon = 1$ and without baseline, and observed the same effect. These empirical results reinforce the theoretical findings of Tamar et al. (2015).

## A.7 ALTERNATE POLICY GRADIENT SUBROUTINES FOR BATCHPOLOPT

As emphasized previously, EPOpt is a generic policy gradient based meta algorithm for finding robust policies. The BatchPolOpt step (line 9, Algorithm 1) calls one gradient step of a policy gradient method, the choice of which is largely orthogonal to the main contributions of this paper. For the

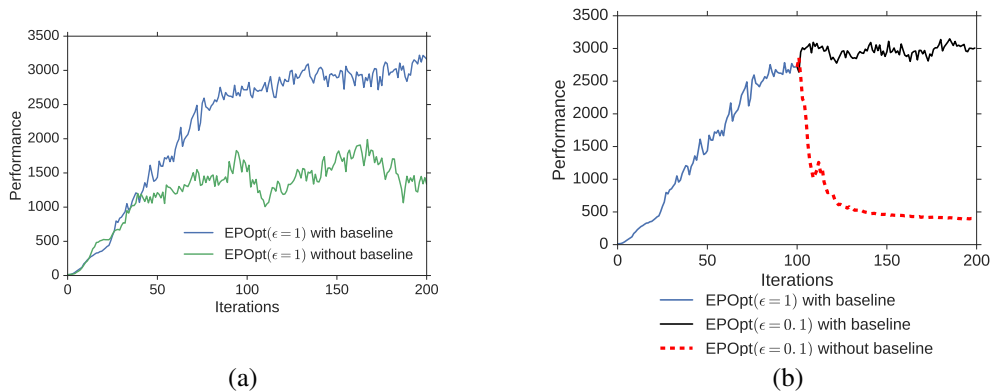

(a)                                                      (b)

Figure 8: (a) depicts the learning curve for EPOpt($\epsilon = 1$) with and without baselines. The learning curves indicate that use of a baseline provides a better ascent direction, thereby enabling faster learning. Figure 8(b) depicts the learning curve when using the average return and CVaR objectives. For the comparison, we "pre-train" for 100 iterations with $\epsilon = 1$ setting and using a baseline. The results indicates that a baseline is very important for the CVaR objective ($\epsilon < 1$), without which the performance drops very quickly. Here, performance is the average return in the source distribution.

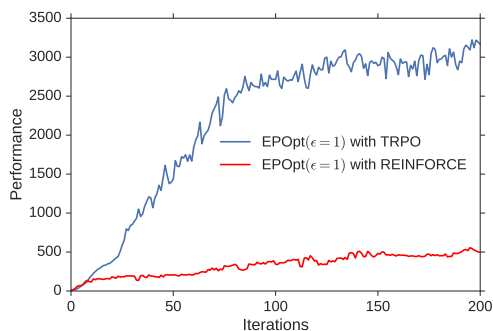

Figure 9: Learning curves for EPOpt($\epsilon = 1$) when using the TRPO and REINFORCE methods for the BatchPolOpt step.

reported results, we have used TRPO as the policy gradient method. Here, we compare the results to the case when using the classic REINFORCE algorithm. For this comparison, we use the same value function baseline parametrization for both TRPO and REINFORCE. Figure 9 depicts the learning curve when using the two policy gradient methods. We observe that performance with TRPO is significantly better. When optimizing over probability distributions, the natural gradient can navigate the warped parameter space better than the "vanilla" gradient. This observation is consistent with the findings of Kakade (2001), Schulman et al. (2015), and Duan et al. (2016).

