# Peer review of "EPOpt: Learning Robust Neural Network Policies Using Model Ensembles"

_ICLR 2017 — accepted_

[Official Review · AnonReviewer2 · rating 7 · confidence 4 · 16 Dec 2016]
**ICLR 2017 conference review**
originality 3 · clarity 2 · impact 1 · substance 2 · meaningful comparison 2

Paper addresses systematic discrepancies between simulated and real-world policy control domains. Proposed method contains two ideas: 1) training on an ensemble of models in an adversarial fashion to learn policies that are robust to errors and 2) adaptation of the source domain ensemble using data from a (real-world) target domain. 

> Significance

Paper addresses and important and significant problem. The approach taken in addressing it is also interesting 

> Clarity

Paper is well written, but does require domain knowledge to understand. 

My main concerns were well addressed by the rebuttal and corresponding revisions to the paper.

[Reviewer Comment · AnonReviewer1 · rating 7 · 16 Dec 2016]
soundness 3 · originality 4 · substance 3

The paper looks at the problem of transferring a policy learned in a simulator to  a target real-world system.  The proposed approach considers using an ensemble of simulated source domains, along with adversarial training, to learn a robust policy that is able to generalize to several target domains.

Overall, the paper tackles an interesting problem, and provides a reasonable solution.  The notion of adversarial training used here does not seem the same as other recent literature (e.g. on GANs).  It would be useful to add more details on a few components, as discussed in the question/response round.  I also encourage including the results with alternative policy gradient subroutines, even if they don’t perform well (e.g. Reinforce), as well as results with and without the baseline on the value function. Such results are very useful to other researchers.

[Official Review · AnonReviewer3 · rating 8 · confidence 4 · 19 Dec 2016 (modified: 20 Dec 2016)]
**Ensemble training and transfer, a good submission**
soundness 3 · originality 3 · impact 2 · substance 4

This paper explores ensemble optimisation in the context of policy-gradient training. Ensemble training has been a low-hanging fruit for many years in the this space and this paper finally touches on this interesting subject. The paper is well written and accessible. In particular the questions posed in section 4 are well posed and interesting.

That said the paper does have some very weak points, most obviously that all of its results are for a very particular choice of domain+parameters. I eagerly look forward to the journal version where these experiments are repeated for all sorts of source domain/target domain/parameter combinations.

[Final Decision · Program Chairs · 06 Feb 2017]
**ICLR committee final decision**

The approach here looks at learning policies that are robust over a parameterized class of MDPs (in the sense the probability that the policy doesn't perform well is small over this class. The idea is fairly straightforward, but the algorithm seems novel and the results show that the approach does seem to provide substantial benefit. The reviewers were all in agreement that the paper is worth accepting.
 
 Pros:
 + Nice application of robust (really stochastic, since these are chance constraints) optimization to policy search
 + Compelling demonstration of the improved range of good performance over methods like vanilla TRPO
 
 Cons:
 - The question of how to parameterize a class of MDPs for real-world scenarios is still somewhat unclear
 - The description of the method as optimizing CVaR seems incorrect, since they appear to be using an actual chance constraint, whereas CVaR is essentially a convex relaxation ... this may be related to the work in (Tamar, 2015), but needs to be better explained if so.